# An Oral Botanical Supplement Improves Small Intestinal Bacterial Overgrowth (SIBO) and Facial Redness: Results of an Open-Label Clinical Study

**DOI:** 10.3390/nu16183149

**Published:** 2024-09-18

**Authors:** Mildred Min, Dawnica Nadora, Mincy Chakkalakal, Nasima Afzal, Chaitra Subramanyam, Nimrit Gahoonia, Adrianne Pan, Shivani Thacker, Yvonne Nong, Cindy J. Chambers, Raja K. Sivamani

**Affiliations:** 1Integrative Skin Science and Research, Sacramento, CA 95815, USA; 2College of Medicine, California Northstate University, Elk Grove, CA 95757, USA; 3College of Osteopathic Medicine, Western University of Health Sciences, Pomona, CA 91766, USA; 4College of Osteopathic Medicine, Touro University, Vallejo, CA 94592, USA; 5College of Human Medicine, Michigan State University, Grand Rapids, MI 49503, USA; 6Pacific Skin Institute, Sacramento, CA 95815, USA; 7Department of Dermatology, University of California-Davis, Sacramento, CA 95816, USA

**Keywords:** gut microbiome, rosacea, SIBO, facial erythema, gut–skin axis, botanical

## Abstract

Background: Small intestinal bacterial overgrowth (SIBO) is a common, yet underdiagnosed, gut condition caused by gut dysbiosis. A previous study has shown the potential of herbal therapy, providing equivalent results to rifaximin. Objectives: The objective of this study was to assess how the use of an oral botanical regimen may modulate the gut microbiome, facial erythema, and intestinal permeability in those with SIBO. Methods: This was an open-label prospective study of adults that had lactulose breath test-confirmed SIBO. Participants received a 10-week oral supplementation of a Biocidin liquid tincture and GI Detox+. If participants were found to be non-responsive to treatment after 10 weeks with a persistently positive lactulose breath test, a third oral supplement, Olivirex, was administered for an additional 4 weeks. Lactulose breath tests were administered at baseline, weeks 6, 10, and 14 to assess for SIBO status. A high-resolution photographic analysis system was utilized to analyze changes in facial erythema. Stool sample collections and venipuncture were performed to analyze the gut microbiome and intestinal permeability. Results: A total of 33 subjects were screened with breath testing, and 19 subjects were found to have SIBO. Three of the subjects withdrew during the screening period prior to baseline, and sixteen subjects enrolled. Four subjects dropped out after baseline. Hydrogen-dominant SIBO was the most common subtype of SIBO, followed by methane and hydrogen sulfide. The botanical regimen was most effective for hydrogen- and hydrogen sulfide-dominant SIBO, leading to negative breath test results at week 10 in 42.8% and 66.7% of participants, respectively. Compared to baseline, supplementation with the botanical regimen led to positive shifts in short-chain fatty acid-producing bacteria such as *A. muciniphila*, *F. prausnitzii*, *C. eutectus*, and *R. faecis* by 31.4%, 35.4%, 24.8%, and 48.7% percent at week 10, respectively. The mean abundance of *Firmicutes* decreased by 20.2%, *Bacteroides* increased by 30%, and the F/B ratio decreased by 25.4% at week 10 compared to baseline. At week 10, there was a trending 116% increase in plasma LPS/IgG (*p* = 0.08). There were no significant changes in plasma zonulin, DAO, histamine, DAO/histamine, LPS/IgG, LPS/IgA, or LPS/IgM. Facial erythema was not statistically different at week 6, but at week 10, there was a 20% decrease (*p* = 0.001) in redness intensity. Among the patients that extended to week 14, there was no statistical change in erythema. Conclusions: Supplementation with an antimicrobial botanical supplemental regimen may have therapeutic potential in hydrogen and hydrogen-sulfide subtypes of SIBO. Furthermore, the botanical supplemental regimen may reduce facial erythema, increase SCFA-producing bacteria, decrease the F/B ratio, and modulate markers of intestinal permeability.

## 1. Introduction

Small intestinal bacterial overgrowth (SIBO) is a condition characterized by an increased colonization of pathogenic bacteria in the small intestine [1]. The gut microbiome is a key player in systemic health, and any shifts in its location or composition can lead to dysregulation. The typical location of the majority of the gut microorganisms is thought to be in the distal small intestine and the large intestine; however, extension into the mid and proximal small intestine leads to the development of SIBO [1]. This condition leads to a multitude of symptoms including bloating, gas, abdominal discomfort, excessive flatulence, and malnutrition [2]. These non-specific symptoms have led the condition to be underdiagnosed and mistaken for other conditions.

The bacterial overgrowth of commensal *Streptococcus*, *Staphylococcus*, *Bacteroides*, and *Lactobacillus* are most noted in SIBO. Among pathogenic genera, *Escherichia*, *Klebsiella*, and *Proteus* genera are noted [3]. The overgrowth in pathogenic bacteria is concerning, given the risk of infection. However, pathogenic bacteria are problematic for several reasons. The overgrowth of pathogenic bacteria disrupts normal digestive and absorptive functions, leading to the aforementioned gastrointestinal symptoms. Pathogenic bacteria can produce toxins and harmful metabolites, such as ammonia and hydrogen sulfide, thereby irritating the gut lining and contributing to inflammation. These bacteria can also interfere with the metabolism of bile acids, leading to deficiencies in fat-soluble vitamins, increases in intestinal permeability, and changes in immune responses [3].

Another significant etiological cause of SIBO is dysmotility within the gastrointestinal tract. The migrating motor complex (MMC) is partially responsible for the clearance of debris within the bowel [4]. Conditions that interfere with the MMC including scleroderma, diabetic gastroparesis, and diverticulosis can increase the likelihood of developing SIBO due to excessive stasis [4]. Anatomical abnormalities such as surgical blind loops and obstructions can also contribute to the proliferation of bacteria in the small intestine. The long-term use of proton pump inhibitors has also been implicated in causing SIBO due to decreased gastric acid secretion [4]. The literature has also outlined many other conditions including chronic pancreatic insufficiency, irritable bowel syndrome, Crohn’s disease, and liver cirrhosis; however, the exact role of these conditions leading to the development of SIBO is not fully understood [4,5]. The exact prevalence of SIBO in the general population is unknown; however, studies estimate it to be around 2–22%, making it a relatively common condition [6]. SIBO is standardly diagnosed with a lactulose or glucose breath test and treated with antibiotics such as rifaximin [1]. Furthermore, SIBO can be divided into hydrogen-dominant SIBO, methane-dominant SIBO, and hydrogen sulfide-dominant SIBO based on breath test abnormalities [7]. Hydrogen- and hydrogen sulfide-dominant SIBO are most commonly associated with diarrhea, gas, and bloating, whereas methanogenic SIBO is often associated with stasis and constipation [8]. Furthermore, hydrogen methane-dominant SIBO may present with a wide range of symptoms including general discomfort, fatigue, and reflux [8].

The conventional treatment for SIBO is with antibiotics, which carry limitations. Many insurance plans do not routinely cover rifaximin for the treatment of SIBO as it is not an FDA-approved treatment and is more costly than other antibiotics [9]. Additionally, alternative antibiotics are notorious for not only eradicating pathogenic bacteria but beneficial bacteria as well, which may lead to further complications in establishing a healthy gut microbiome in patients with SIBO. These factors have prompted the investigation for treatment alternatives, and a study conducted in 2014 found herbal therapy to have equivalent results when compared to rifaximin in treating SIBO [9].

Many herbs have been used traditionally to treat bacterial and fungal infections due to their antimicrobial properties. For example, garlic (*Allium sativum*) is a common herb containing allicin, an active compound with antibacterial, antifungal, antiviral, and antiparasitic activity via its reaction with thiol groups of various enzymes [10]. Basil (*Ocimum basilicum*), oregano (*Origanum vulgare*), and thyme (*Thymus vulgaris*) essential oils have also been shown to exhibit antimicrobial activity via mechanisms including hydrophobicity, the impairment of cytoplasmic membranes, disrupting electron flow, and the coagulation of cell contents [11]. Furthermore, olive leaf extract (*Olea europaea*) has been shown to inhibit the biofilm formation of multi-drug-resistant organisms such as *Pseudomonas aeruginosa* [12]. Herbs have also been shown to influence gastrointestinal function. For example, a liquid formulation of different herbs including licorice root (*Glycyrrhiza glabra*), milk thistle (*Silybum marianum*), and peppermint oil (*Mentha piperita*) was found to have antispasmodic, anti-inflammatory, and antioxidant activity, in addition to acting as a secretagogue [13]. Other herbs like fennel (*Foeniculum vulgare*), lavender (*Lavandula angustifolia*), rosemary (*Salvia Rosmarinus*), and sage (*Salvia officinalis*) have been shown to improve digestion and reduce flatulence [14]. Thus, herbal therapies may target several factors contributing to SIBO including bacterial overgrowth and dysmotility. To further investigate the role of herbal therapy in SIBO, we conducted an open-label prospective study to determine the effects of a botanical blend on the gut and skin in patients with SIBO.

## 2. Material and Methods

### 2.1. Materials

The products used in this study were provided by Biocidin Botanicals^®^ (Watsonville, CA, USA). The GI Detox^®^+ capsule contained 520 mg of an herbal blend composed of the following ingredients: zeolite clay, activated charcoal, aloe vera gel, apple pectin, silica, humic acid, and fulvic acid. The Biocidin^®^ liquid tincture (BLT) contained 30 mg of an herbal blend containing the following ingredients: vegetable glycerin, bilberry fruit extract (*Vaccinium myrtillus*), grape seed extract (*Vitis vinifera*), shiitake mushroom extract (*Lentinula edodes*), goldenseal root (*Hydrastis canadensis*), noni fruit extract (*Morinda citrifolia*), garlic bulb (*Allium sativum*), white willow bark (*Salix alba*), milk thistle seed (*Silybum marianum*), echinacea purpurea herb extract (*Echinacea purpurea*), echinacea angustifolia root (*Echinacea angustifolia*), raspberry fruit (*Rubus idaeus*), black walnut hull (*Juglans nigra*), black walnut leaf (*Juglans nigra*), lavender oil (*Lavandula angustifolia*), oregano oil (*Origanum vulgare*), galbanum oil (*Ferula gummosa*), tea tree oil (*Melaleuca alternifolia*), fumitory aerial parts extract (*Fumaria officinalis*), and gentian lutea root (*Gentiana lutea*). The Olivirex^®^ capsule contained 375 mg of olive leaf extract (*Olea europaea*) and 53 mg of an herbal blend containing the following ingredients: garlic bulb (*Allium sativum*), noni fruit extract (*Morinda citrifolia*), uva ursi leaf (*Arctostaphylos uva-ursi*), milk thistle seed extract (*Silybum marianum*), cordyceps fruiting body (*Cordyceps sinensis*), St. John’s Wort aerial parts (*Hypericum perforatum*), dandelion root (*Taraxacum officinale*), goldenseal root (*Hydrastis canadensis*), white willow bark (*Salix alba*), bladderwrack thallus (*Fucus vesiculosus*), and American ginseng root extract (*Panax quinquefolius*). Study products were stored in a secured area at room temperature (20–22 °C). Participants were instructed to keep the study products in a cool, dry place, as per the suggested storage instructions.

### 2.2. Supplemental Regimen

Signed informed consent was obtained for all participants prior to enrollment. Subjects who qualified for the study were given oral supplementations of GI Detox+ capsules and the Biocidin liquid tincture (BLT) daily for 10 weeks. Subjects were instructed to take 1 drop of the BLT orally twice daily, increasing by 1 drop each day until 15 drops twice daily were reached. Fifteen drops of the BLT twice daily were to be maintained throughout the duration of the study. Subjects were also instructed to take 2 GI Detox+ capsules orally every evening throughout the duration of the study. Subjects who tested negative for SIBO at 10 weeks were labeled as “responders” and terminated the study at 10 weeks. Subjects who tested positive for SIBO at 10 weeks were labeled as “non-responders”. Non-responders received a third oral supplement, Olivirex, in addition to the current regimen for an additional 4 weeks. Two Olivirex capsules were to be taken orally twice daily with the BLT. Non-responders concluded study participation at 14 weeks. Table 1 demonstrates the regimen that participants were instructed to follow for the duration of the study.

### 2.3. Study Design and Recruitment

This study was an open-label prospective clinical trial conducted from April 2021 to April 2023. The study was reviewed and approved by the Integreview Institutional Review Board on 5 March 2021 (Protocol ID: BIOSIBO1) and registered at www.clinicaltrials.gov (accessed on 24 September 2023, NCT04867512). All subjects were recruited from the greater Sacramento region and completed the study visits at Integrative Skin Science and Research in Sacramento, CA. The study visits included screening, baseline, 6, 10, and 14 weeks. All enrolled participants received the GI Detox+ capsule and BLT for 10 weeks. If participants were found to be non-responsive to treatment after 10 weeks on a lactulose breath test, a third oral supplement, Olivirex, was added to the regimen for an additional 4 weeks. Participants were instructed to record their use of the supplements as part of the study’s monitoring methods, and coordinators reviewed this at each visit.

Of the 113 individuals assessed for eligibility, 19 met all the inclusion criteria and none of the exclusion criteria, and 16 of these individuals were fully enrolled in the study. Any adverse events of the intervention were noted throughout the study. A CONSORT diagram is presented in Figure 1.

### 2.4. Subject Inclusion and Exclusion Criteria

Subjects included males and females over the age of 18 years with SIBO as diagnosed by a lactulose breath test. Subjects who had recently used oral or injected antibiotics within the last 6 months, topical antibiotics or benzoyl peroxide within the last 2 months, and oral probiotic or prebiotic use within the last 1 month were excluded from the study. Additionally, subjects with a BMI greater than 35 kg/m^2^, nut allergy, use of statins or any other anti-hyperlipidemic medications, history of harmful alcohol use as classified by the Alcohol Use Disorder Identification Test, and commencement of a new diet within the last 1 month were excluded. Subjects were also not enrolled into the study if they had a previous medical history of any epilepsy, cancer (excluding non-malignant skin cancer), gastrointestinal, immunologic, or infectious diseases. Current smokers, or those who have smoked in the past year, or had a 5-year pack-year history within the past 10 years were also excluded.

### 2.5. Lactulose Breath Testing

Lactulose breath tests were administered to determine positive or negative SIBO status at baseline, week 6, 10, and 14. Participants utilized the breath collection and testing kit by Trio-Smart (Gemelli Biotech, Raleigh, NC, USA). A breath sample was collected at baseline prior to the ingestion of 10 g of lactulose in 15 mL of solution and then every 15 min for a total of 120 min after ingestion. Tests were considered positive if there was (1) a breath concentration of ≥20 ppm for hydrogen, (2) ≥10 ppm for methane, or (3) ≥3 ppm for hydrogen sulfide within the 120 min testing window after lactulose ingestion.

### 2.6. Facial Imaging and Image-Based Analysis

The BTBP 3D Clarity Pro Facial Modeling and Analysis System (Brigh-Tex BioPhotonics, San Jose, CA, USA) was utilized to capture high-resolution facial images for all study participants at baseline, 6, 10, and 14 weeks. The facial redness intensity was utilized to measure facial redness based on previous work that has validated this approach for accurate facial redness assessment [15].

### 2.7. Gut Microbiome Sequencing and Analysis

Stool sample collection kits were distributed to participants at baseline and week 6 in both groups, with an additional kit distributed for non-responders at week 10. Participants received detailed instructions on stool sample collection and mailed out their stool samples to Sun Genomics (Sun Genomics Inc., San Diego, CA, USA) for sequencing and analysis. Stool samples were analyzed at baseline, 10, and 14 weeks (non-responders only).

For DNA extractions, samples were first processed with a tissue homogenizer and then lysed with a lysis buffer and proteinase K. DNA was extracted and purified with a proprietary method (patents 10,428,370 and 10837046). Library preparation was performed with DNA sonication, end-repair, and adaptor ligation with NEBNext (NEB, Ipswich, MA, USA) reagents. Size selection was performed with MagJet (Thermo Fisher Scientific, Waltham, MA, USA) magnetic beads according to the manufacturer’s instructions. Library quantitation was performed with quantitative PCR (qPCR), and sequencing was performed with an Illumina NextSeq 550 (Illumina, San Diego, CA, USA). After sequencing, the reads were quality-filtered and processed. Metagenomic reads were decontaminated from human reads using Bowtie2. After decontamination, there was an average of 6,581,844 reads per sample (SD = 4,426,117), with a minimum of 1 million reads to be included in downstream analyses. Next, reads were aligned to a hand-curated database of over 23,000 species. Humann3 was used for pathway analysis. Pathway abundance was normalized to copies per million (cpm).

### 2.8. Plasma Collection and Intestinal Barrier Assessment

Venipuncture was performed at baseline, 10, and 14 weeks (non-responders only). Plasma samples were sent to Precision Point Diagnostics (Precision Point Diagnostics Inc., Dunwoody, GA, USA) for the quantification of intestinal barrier markers. Intestinal barrier markers of interest included zonulin, diamine oxidase (DAO), histamine, DAO/histamine, lipopolysaccharide (LPS)/immunoglobin A (IgA), LPS/IgG, and LPS/IgM, and these were measured in plasma. All biomarkers were quantified utilizing enzyme-linked immunosorbent assays (ELISA) following the manufacturer’s protocols.

### 2.9. Statistical Analysis

The data were analyzed at 0, 6, 10, and 14 weeks. Statistical comparisons were made using a Student’s t-test for differences (paired, two-tailed) with statistical significance set at *p* ≤ 0.05. Results are presented as the mean with standard error of mean. Baseline data from all subjects were utilized as controls for within-group comparisons as each participant served as their own control. Prism V.10 (GraphPad Software LLC, San Diego, CA, USA) was used to analyze and visualize the data.

## 3. Results

Nineteen participants met all the inclusion and exclusion criteria for this study, and sixteen were fully enrolled to receive the open-label intervention. The mean (SD) age for participants was 44.7 ± 14.4 y. There were seventeen females (89.4%) and two males (10.6%) enrolled.

### 3.1. Lactulose Breath Test Results

Five out of nineteen participants tested positive for more than one subtype of SIBO at screening, and sixteen participants with positive SIBO breath tests completed baseline (Table 2). Thirteen participants tested positive for hydrogen SIBO at baseline, and 37.5.% (*n* = 3) of participants at week 6 (*n* = 8) were hydrogen SIBO-negative. By week 10, 66.7% (*n* = 4) of participants (*n* = 6) were hydrogen SIBO-negative. None of the non-responders became hydrogen SIBO-negative at week 14. Seven participants tested positive for methanogenic SIBO at baseline, and none were methane SIBO negative at week 6. There were no methanogenic SIBO responders at week 10; however, 25% (*n* = 1) of participants (*n* = 4) were methane SIBO-negative at week 14. There were four participants that tested positive for hydrogen sulfide SIBO at baseline, and 100% (*n* = 4) of participants (*n* = 4) were hydrogen sulfide SIBO-negative at week 6. Of the three remaining participants at week 10 and 14, 66.7% (*n* = 2) were hydrogen sulfide SIBO-negative at week 10 and 100% (*n* = 3) were hydrogen sulfide SIBO-negative at week 14. The lactulose breath test results for subjects at baseline, weeks 6, 10, and 14 are summarized in Table 3 and Figure 2.

### 3.2. Facial Erythema

The facial erythema intensity was evaluated per protocol in participants that completed baseline, week 6, 10, or 14, respectively. Facial erythema intensity was not statistically different at 6 weeks (6.9 ± 12.2%, *p* = 0.58) and was reduced at week 10 (−20.0 ± 4.8%, *p* = 0.001) compared to baseline. The facial erythema intensity in pure H2-dominant SIBO was not statistically different at 6 weeks (18.4 ± 27.4%, *p* = 0.53) and was reduced at week 10 (−14.4 ± 3.9%, *p* = 0.013) compared to baseline. The facial erythema intensity in pure CH4-dominant SIBO was not statistically different at 6 weeks (−1.5 ± 16.7%, *p* = 0.93) and was reduced at week 10 (−30.7 ± 7.3%, *p* = 0.014) compared to baseline. The facial erythema intensity in H2-dominant SIBO (mixed included) was not statistically different at 6 weeks (8.6 ± 18.4%, *p* = 0.65) and was reduced at week 10 (−16.2 ± 6.2%, *p* = 0.03) compared to baseline. The facial erythema intensity in CH4-dominant SIBO (mixed included) was not statistically different at 6 weeks (0.2 ± 13.2%, *p* = 0.99) and was reduced at week 10 (−25.6 ± 6.5%, *p* = 0.008) compared to baseline. The facial erythema intensity in H2S-dominant SIBO (mixed included) was not statistically different at 6 weeks (8.4 ± 12.8%, *p* = 0.58) and was not statistically different at week 10 (−12.1 ± 199%, *p* = 0.61) compared to baseline. These results are shown in Figure 3.

High-resolution photography was taken for all participants at baseline, week 6, and week 10, with an additional photo at week 14 for non-responders (Figure 4).

### 3.3. Gut Microbiome Findings

There was a mean 31.4% increase in the relative abundance of *Akkermansia muciniphila*, 35.4% increase in *Faecalibacterium prausnitzii*, 24.8% increase in *Coprococcus eutectus*, and 48.7% increase in *Roseburia faecis* at week 10 compared to baseline (Figure 5). There was a mean 9.8% increase in the relative abundance of *Akkermansia muciniphila*, 58.5% increase in *Faecalibacterium prausnitzii*, 18.2% decrease in *Coprococcus eutectus*, and 26.3% decrease in *Roseburia faecis* at week 14 compared to baseline (Appendix A). None of these changes were statistically significant.

The mean abundance of *Firmicutes* decreased by 20.2% from baseline (mean, 16.9) to week 10 (mean, 13.5). There was a 30% increase in *Bacteroides* from baseline (mean, 27.5) to week 10 (mean, 35.7). The F/B ratio decreased by 25.4% from baseline (mean, 0.94) to week 10 (mean, 0.70). These findings can be seen in Figure 6. The mean abundance of *Firmicutes* decreased by 29.5% from baseline (mean, 16.9) to week 14 (mean, 11.9), there was a 46.1% decrease in *Bacteroides* from baseline (mean, 27.5) to week 14 (mean, 14.8), and the F/B ratio increased by 52.9% from baseline (mean, 0.94) to week 14 (mean, 1.4) (Appendix A).

### 3.4. Plasma Intestinal Barrier Function Markers

Mean plasma zonulin levels decreased in all participants after 10 weeks of supplementation. After 10 weeks of the botanical regimen, participants had a statistically significant 26% increase in plasma DAO (*p* = 0.05) (Figure 7). There were no significant changes in plasma histamine, DAO/histamine, LPS/IgA, LPS/IgG, and LPS/IgM levels after 10 weeks. At week 14, there were no statistically significant differences (Appendix A).

Among participants with hydrogen-dominant SIBO, there were no significant changes in any of the intestinal barrier markers at 10 weeks (Figure 8). Among participants with methanogenic SIBO, there was an 88.8% decrease in histamine at week 10 (*p* = 0.04) (Figure 8). Among participants with multiple subtypes, there were no significant changes at 10 weeks (Figure 8). There were no significant changes at week 14 for any of the cohorts (Appendix A).

### 3.5. Adverse Events

There was one participant that dropped out of the study after their baseline visit due to headaches. The headaches were determined to not be related to investigational product use as they were persistent after the discontinuation of the study intervention.

## 4. Discussion

This study found that the use of a botanical supplemental regimen in SIBO patients may mitigate certain subtypes of SIBO, reduce facial erythema, lead to beneficial shifts in gut microbiota, and influence intestinal barrier function. Although a previous study has shown that herbal therapy is as effective as rifaximin [9], our study is unique in that we extend our analysis to correlate SIBO disease status with facial erythema, the gut microbiome, and intestinal permeability.

### 4.1. Lactulose Breath Test Findings

Our study found that the botanical supplemental regimen provided had inconclusive effects on methanogenic SIBO. The BLT provided to participants in this study contained grape seed extract (*Vitis vinifera*), bilberry fruit extract (*Vaccinium myrtillus*), and garlic bulb (*Allium sativum*). Garlic bulbs are rich in allicin, a compound that has been shown to limit methane-producing microbes. For example, one study found that allicin supplementation in sheep reduced daily methane emissions; however, the mechanism by which this occurs remains unknown [16]. Similarly, an in vitro study performed on rumen fluid demonstrated that grape seed extract and bilberry fruit extract reduced the production of methane [17]. Although the ingredients making up the BLT have promising potential for the treatment of methanogenic SIBO, our study did not have similar findings, and more research is needed to expand on these preliminary in vivo findings.

Interestingly, the variable increases seen in methane production may be due to the zeolite clay or activated charcoal found in the GI Detox+ capsules. One study conducted on various types of zeolites on biogas production found that the *Bacteroides* phylum is mainly responsible for the degradation of complex organic compounds, so the presence of this phylum contributes to high methane production [18]. Thus, the increased abundance in *Bacteroides*, as shown in our study, may explain the sustained methane production seen in our methanogenic SIBO participants. Additionally, the activated charcoal found in the capsules may have also contributed to the increase in methane seen in the participants. Another study reported that activated carbon/graphite enhanced the anaerobic digestion of waste-activated sludge and reported that methane production is increased by promoting the consumption of hydrolysis and acidification products in the methanogenesis process [19].

There were several participants that were cleared of their hydrogen-dominant SIBO by week 6, and this could have been due to ingredients such as oregano (*Origanum vulgare*) and berberine found in the BLT, which have been shown to be effective against hydrogen-producing bacteria [20,21]. Furthermore, increases in propionate-producing bacteria, such as *Bacteroides* [22], may inhibit hydrogen-producing bacteria, as shown in our study. Similarly, an in vivo study reported that the phenolics in oregano essential oils inhibit methanogenic and hydrogen-producing bacteria in rumen fermentation [23]. However, the increase in hydrogen gas seen in some participants may be supported by a study that found that increasing the dosage of oregano oil led to a linear increase in hydrogen production in cows’ feed [24]. Additionally, berberine has been shown to enhance the composition of beneficial bacteria such as *Bacteroides*, *Bifidobacterium*, *Lactobacillus*, and *Akkermansia* [25]. Another study found that herbal therapy including berberine and oregano oil appeared to be similarly effective in the treatment of SIBO patients compared to rifaximin [9].

Our study also found that several participants were cleared of their hydrogen-sulfide SIBO by week 6. Although there is limited research on how botanicals can influence hydrogen sulfide production, our findings are nevertheless supported by studies showing that herbal therapeutic regimens have antimicrobial properties that can help mitigate SIBO [26]. This study is the first to note the clearance of hydrogen sulfide SIBO as a result of a botanical intervention.

Interestingly, there were a few participants across all SIBO subtypes that cleared their SIBO and then tested positive again later during the study at week 10 or 14. Some of these participants went on to clear their SIBO again, but a few remained SIBO positive at the end of the study. One reason for this may be due to the complex nature of SIBO, where the colonization of cross-feeders may influence breath test results. For example, in many individuals with methane-positive SIBO, a hydrogen cross-feeder, *Methanobrevibacter smithii*, converts H2 and CO2 into CH4 [8]. More research is needed to understand the different subtypes of SIBO and how clearing one subtype may influence the production of other gasses.

### 4.2. Changes in Facial Erythema

The use of the botanical supplemental regimen led to a significant reduction in facial erythema at week 10 compared to baseline overall. Our results also showed that the improvement is noted whether the subtype of SIBO present was hydrogen-dominant or methane-dominant (Figure 3). While the etiology of these two subtypes of SIBO appear to be a different etiology, treatment with the regimen utilized here appeared to impact the skin regardless of subtype. When including participants that had a mixed presentation of SIBO, the sub-analysis still showed statistically significant improvements in the facial redness in those that had hydrogen-dominant and methane-dominant SIBO. We could not perform any similar conclusions for hydrogen sulfide-dominant SIBO, because there were no subjects that presented with only hydrogen sulfide-dominant SIBO and there were only three subjects that had a mixed presentation with hydrogen sulfide as one of the subtypes. Future studies with expanded populations of the SIBO subtypes would better delineate the effects of supplementation and facial redness. What is particularly notable is that although the supplementation was less efficacious at clearing methane-dominant SIBO, there was still a robust reduction in the facial redness in this subgroup, suggesting that the clearance of methane SIBO may be independent from the facial redness reduction.

Our findings are in agreement with previous work in SIBO and facial redness. The prevalence of SIBO and gut dysbiosis has been shown to be markedly higher in patients with rosacea, a chronic inflammatory condition leading to facial redness, flushing, pustules, and telangiectasias [27]. One study has shown that eradicating SIBO in patients with rosacea with rifaximin leads to a nearly complete regression of their cutaneous lesions [28]. Therefore, the mitigation of certain subtypes of SIBO by the botanical supplemental intervention may have also influenced rosacea severity, thus reducing facial erythema overall. Moreover, the gut microbiome has been shown to influence skin health, and the mechanism by which the study regimen may have reduced facial erythema may be through its impact on the gut microbiome and related biochemical pathways [27,29].

Currently, systemic antibiotics are the typical first-line therapy in the treatment of rosacea. However, there is a need to reduce the use of antibiotics to reduce the development of resistant bacteria [30]. Furthermore, the gut microbiome can be altered for prolonged periods of time after antibiotic exposure [31]. Therapies that serve as alternatives to antibiotics will be important, and our study suggested that the botanical regimen studied here may serve as a suitable alternative in reducing facial erythema. Future studies in a larger cohort of patients with rosacea would be warranted.

### 4.3. Gut Microbiota

The gut microbiome profile of SIBO patients has been shown to be distinct from those of healthy controls [32]. In our study, after 10 weeks of Biocidin liquid tincture and GI Detox+ supplementation, there were increased relative abundances of *Akkermansia muciniphila*, *Faecalibacterium prausnitizii*, *Coprococcus eutectus*, *Roseburia faecis*, and *Bacteroides* when compared to baseline. The existing literature suggests that increased relative abundances of these bacteria have been associated with short-chain fatty acid (SCFA) production, decreased intestinal permeability, metabolism, and more. Thus, these shifts in the gut microbiota may help to mitigate symptoms of SIBO.

For example, *A. muciniphila* is an SCFA producer through the degradation of mucin [33]. SCFAs such as acetic acid, propionic acid, and butyric acid are byproducts of carbohydrate metabolism and act as the main energy source for the intestinal epithelium [34]. Butyrate has been shown to prevent stem cell inhibition, maintain luminal anaerobiosis, which is favorable for commensal bacteria, and stabilize gut barrier function through anti-inflammatory effects [35]. Although *A. muciniphila* has primarily been studied for its pro-metabolic effects [36], an increase in *A. muciniphila* and its consequent increase in SCFAs may modulate SIBO by improving intestinal barrier function, as shown in our study.

*F. prausnitzii* is also an SCFA producer involved in glucose metabolism [37]. Notably, *F. prausnitzii* also produces butyrate, like *A. muciniphila* [38]. *F. prausnitzii* is a commensal bacteria that has been shown to have anti-inflammatory effects through the inhibition of NF-κB activation and IL-8 synthesis [39]. Due to its anti-inflammatory effects, some studies have postulated that the use of *F. prausnitzii* as a probiotic may mitigate gut dysbiosis and have therapeutic applications for irritable bowel syndrome (IBS) or Crohn’s disease [39,40]. Alternatively, some studies have reported contradicting findings for the role of *F. prausnitzii* in gut health. For example, one study found that a significant decrease in the abundance of *F. prausnitzii* was correlated with the clinical improvement of Crohn’s disease [41]. Another study found that SIBO-positive patients had a significantly higher abundance of *F. prausnitzii* when compared to healthy controls [32]. Thus, while the mechanism of action of *F. prausnitzii* suggests that the increased abundance of this bacterium may improve gastrointestinal health, more research is needed to clarify the contradicting results in these studies. Furthermore, *C. eutectus* and *R. faecis* are also SCFA-producing bacteria that may improve SIBO by augmenting intestinal barrier function [42,43].

Interestingly, the relative abundances of *C. eutectus* and *R. faecis* decreased at 14 weeks when compared to baseline. However, the relative abundances of *A. muciniphila* and *F. prausnitizii* remained elevated compared to baseline. It is unclear why *C. eutectus* and *R. faecis* shifted in this manner. This finding may be related to the non-responder status of participants that continue through to week 14, and more research is needed to better understand if and how these microbiota drive treatment-resistant SIBO.

There was a decrease in the *Firmicutes* phylum and an increase in *Bacteroides* at week 10, leading to an overall decreased *Firmicutes*/*Bacteroides* (F/B) ratio. The F/B ratio has been shown to act as a biomarker for inflammation, and an increased F/B ratio is typically indicative of dysbiosis and increased inflammation [44,45]. Thus, the decreased F/B ratio, as seen in our study after the botanical supplemental regimen, supports that the studied blend of herbs may mitigate dysbiosis in SIBO and reduce inflammation. However, at week 14, there was a decrease in the *Firmicutes* phylum and a decrease in *Bacteroides*, with an overall increased *Firmicutes*/*Bacteroides* (F/B) ratio. Similar to the gut microbiota, it is unclear why these shifts occurred after the initial shifts seen at week 10. Again, since only non-responders were participating past week 10, it is possible that there is a gut microbiome-related mechanism that drives treatment-resistant SIBO.

### 4.4. Changes Plasma Intestinal Barrier Function Markers

There were several notable findings in plasma markers for intestinal barrier function. Compared to baseline, there was a decrease in zonulin at week 10, which continued to decrease at week 14. Circulating zonulin is a marker of intestinal permeability [46] and has been shown to modulate intercellular tight junctions in the gut [47]. Elevated levels of circulating zonulin have been associated with numerous chronic inflammatory, autoimmune, and malignant diseases [48]. Therefore, the reduction in plasma zonulin levels, as seen in our study, may support the anti-inflammatory and gut barrier-supporting properties of herbal therapies. Moreover, our study found that there was a significant increase in diamine oxidase (DAO) at week 10. DAO is the primary extracellular enzyme involved in the degradation of histamine and biogenic amines in the gut [49]. When the degradation of biogenic amines by DAO is dysregulated, histamine can accumulate and contribute to various intestinal pathologies [49]. Thus, sufficient levels of circulating DAO are necessary to mitigate histaminergic symptoms such as allergies, digestive issues, and more. Alternatively, DAO has also been shown to increase in response to inflammation or trauma [50]. As a result, it remains unclear whether increased plasma DAO, as seen in our study, may play a role in supporting gastrointestinal health. There were also variable changes in LPS antibodies, with levels generally trending upward. High levels of LPS antibodies have been shown to protect against microbes and maintain intestinal homeostasis [51]. LPS antibodies such as LPS/IgA and LPS/IgG were increased at week 14, and these antibodies may contribute to maintaining luminal barrier function. Otherwise, there were also variable changes in markers such as histamine and DAO/histamine. While histamine has been shown to regulate mucosal immune responses [52], more research is needed to assess how herbal therapies may influence circulating histamine levels. These shifts in plasma intestinal barrier function markers may also correlate with the gut microbiome shifts demonstrated in our study. Positive shifts in SCFA producers promote mucin production, tighten epithelial junctions, and modulate inflammatory immune responses [53]. Subsequently, SCFAs improve intestinal barrier function, and this is reflected in improved plasma markers of intestinal barrier function, such as low levels of zonulin [54]. Similarly, a lower F/B ratio, as previously mentioned, is associated with anti-inflammatory properties, leading to improved barrier function [55].

There were also some variable changes in intestinal barrier biomarkers noted when stratifying the results by different SIBO subtypes. For participants with hydrogen-dominant SIBO, there was a trending increase in LPS/IgG, which follows the general trend seen when analyzing all the participants. However, participants with hydrogen-dominant SIBO experienced increases in zonulin from baseline, which is different from the general trend. Among methanogenic SIBO participants, there was a significant decrease in histamine and a decreasing DAO/histamine, which are both different from the general trend. Yet, the trend in other biomarkers remained relatively similar to the overall analysis. In participants with multiple subtypes of SIBO, there were no notable changes when compared to the analysis of all participants. To the best of our knowledge, no studies have investigated how the SIBO subtype may influence intestinal barrier biomarkers, and more research is needed to understand how these differences may contribute to disease severity and persistence.

### 4.5. Limitations

The sensitivity of the breath test used to assess the presence of SIBO has some limitations, although it is still the best approach to testing without invasive testing, such as gastrointestinal aspirates. Moreover, the breath test used was the most robust currently available on the market and the only one that included hydrogen sulfide assessment. This study serves as a pilot study with a limited number of participants. There was also a relatively high dropout rate in this study; however, several findings were still significant and should prompt further and larger clinical studies with extended microbiome analysis. Nevertheless, this study serves as a pilot study to reveal potential mechanisms by which herbal therapy may mitigate SIBO. Most of the participants in this study were female, and the results may not be extended to males without future studies with more male recruitment. Furthermore, the cost of the herbal regimen is typically not covered by commercial insurance. However, it is relatively affordable, and individuals may be able to use their flexible spending accounts to cover the expenses.

## 5. Conclusions

The data suggest that a botanical supplemental regimen including GI Detox+ and the Biocidin liquid tincture may reduce facial erythema, influence shifts in beneficial gut bacteria that contribute to barrier function, and regulate intestinal permeability. Future prospective studies that include expanded populations and a comparison against antibiotics would be warranted.

## Figures and Tables

**Figure 1 nutrients-16-03149-f001:**
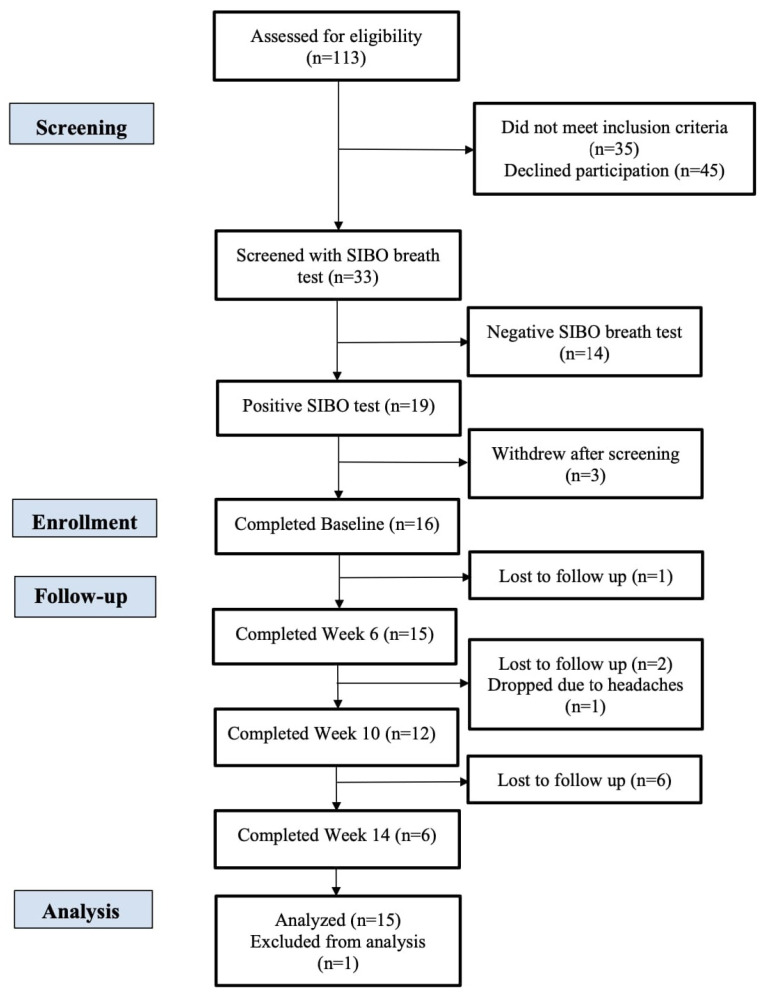
CONSORT (Consolidated Standards of Reporting Trials) flow diagram.

**Figure 2 nutrients-16-03149-f002:**
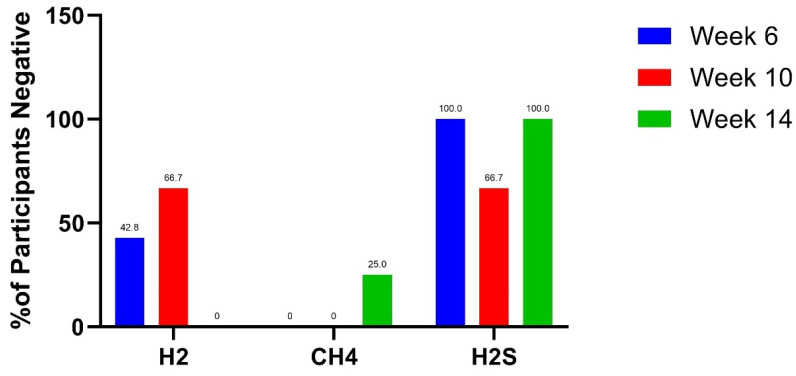
Proportion of participants with negative breath test results for H2 (hydrogen), CH4 (methane), or H2S (hydrogen sulfide) SIBO at week 6, 10, and 14 after initially testing positive at baseline.

**Figure 3 nutrients-16-03149-f003:**
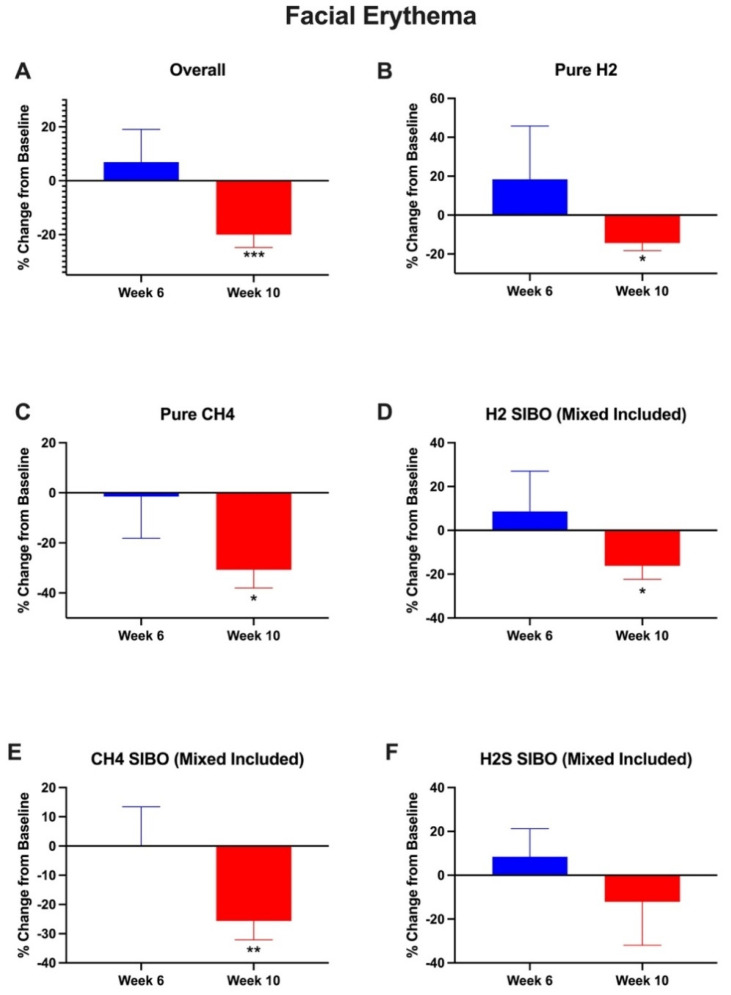
Facial erythema intensity among all participants at week 6 and week 10. Image-based photographic analysis of erythema intensity was significantly decreased in (**A**) all participants, (**B**) H2-dominant SIBO, (**C**) CH4-dominant SIBO, (**D**) H2-dominant SIBO (mixed included), (**E**) CH4-dominant SIBO (mixed included), (**F**) H2S-dominant SIBO (mixed included) at week 10. Error bars represent the SEM. * = *p* < 0.05. ** = *p* < 0.01. *** = *p* < 0.001.

**Figure 4 nutrients-16-03149-f004:**
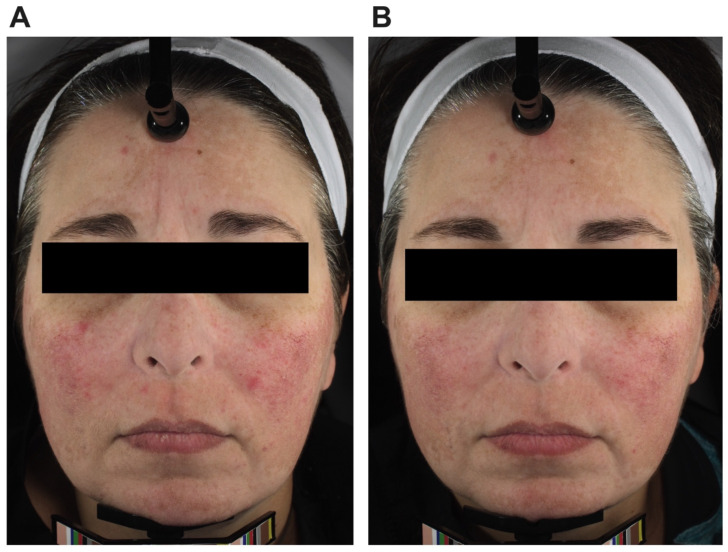
High-resolution facial images taken at (**A**) baseline and (**B**) week 10 showing reduced facial erythema.

**Figure 5 nutrients-16-03149-f005:**
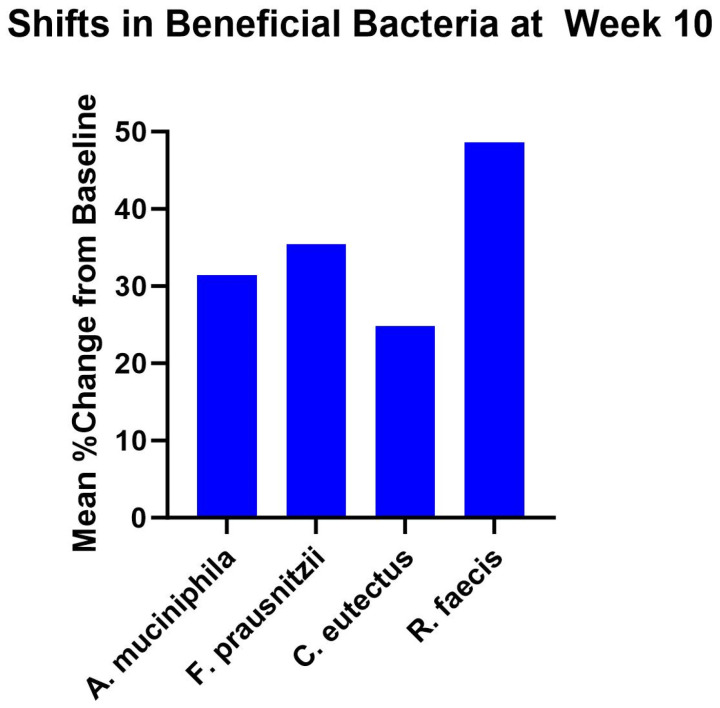
Gut microbiome analysis of beneficial bacteria demonstrated shifts in *Akkermansia muciniphila*, *Faecalibacterium prausnitzii*, *Coprococcus eutectus*, *and Roseburia faecis* at week 10 compared to baseline.

**Figure 6 nutrients-16-03149-f006:**
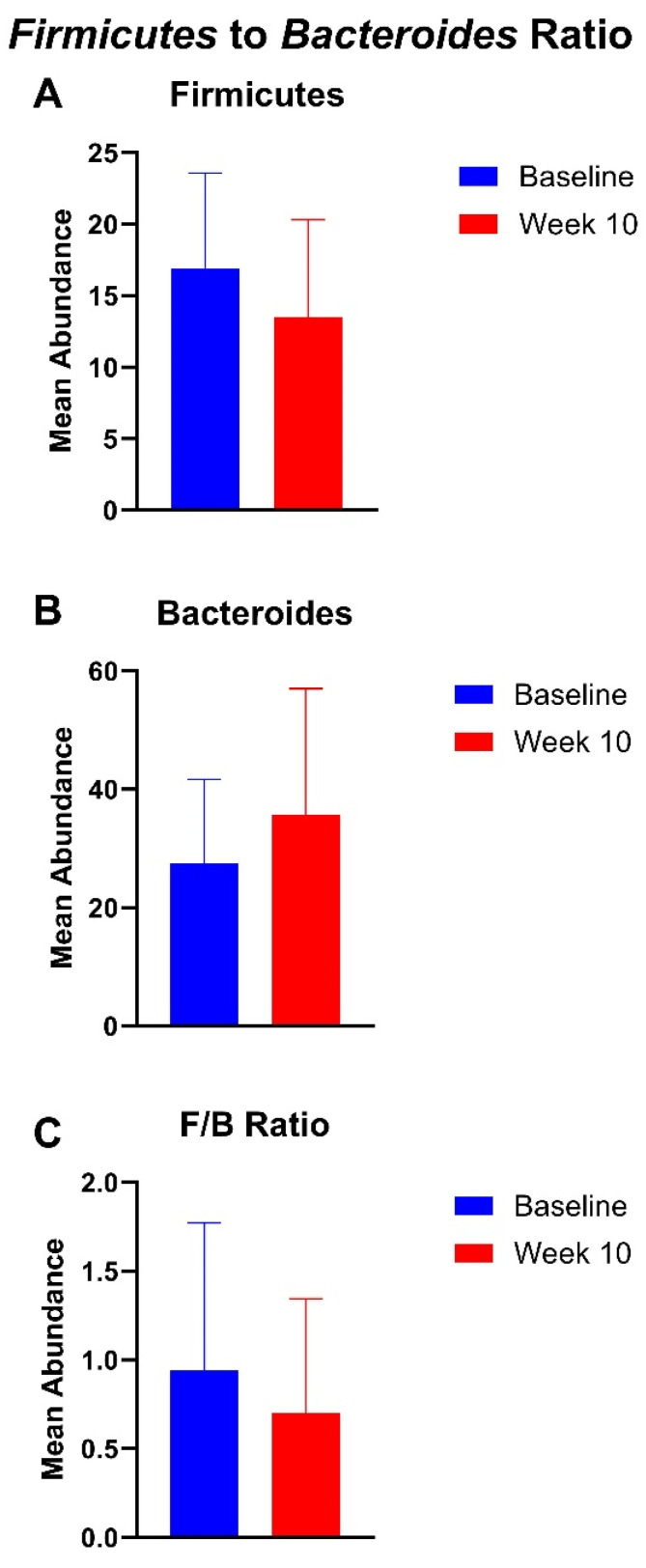
Gut microbiome analysis of *Firmicutes* and *Bacteroides* demonstrated decreased (**A**) mean abundance of Firmicutes, (**B**) increased abundance of Bacteroides, and (**C**) decreased F/B ratio at week 10 compared to baseline. Error bars represent standard deviation (SD).

**Figure 7 nutrients-16-03149-f007:**
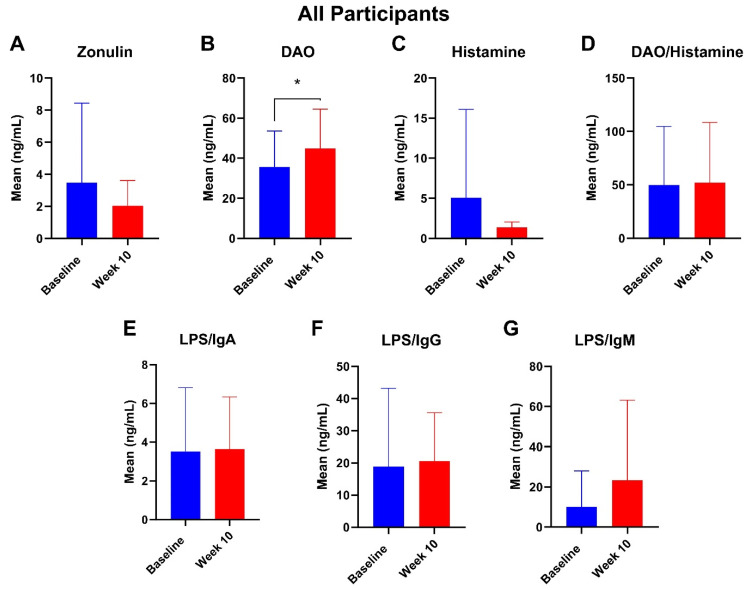
Mean plasma intestinal barrier function markers (**A**) zonulin, (**B**) DAO, (**C**) histamine, (**D**) DAO/histamine, (**E**) LPS/IgA, (**F**) LPS/IgG, and (**G**) LPS/IgM for all participants at baseline and week 10. * = *p* ≤ 0.05.

**Figure 8 nutrients-16-03149-f008:**
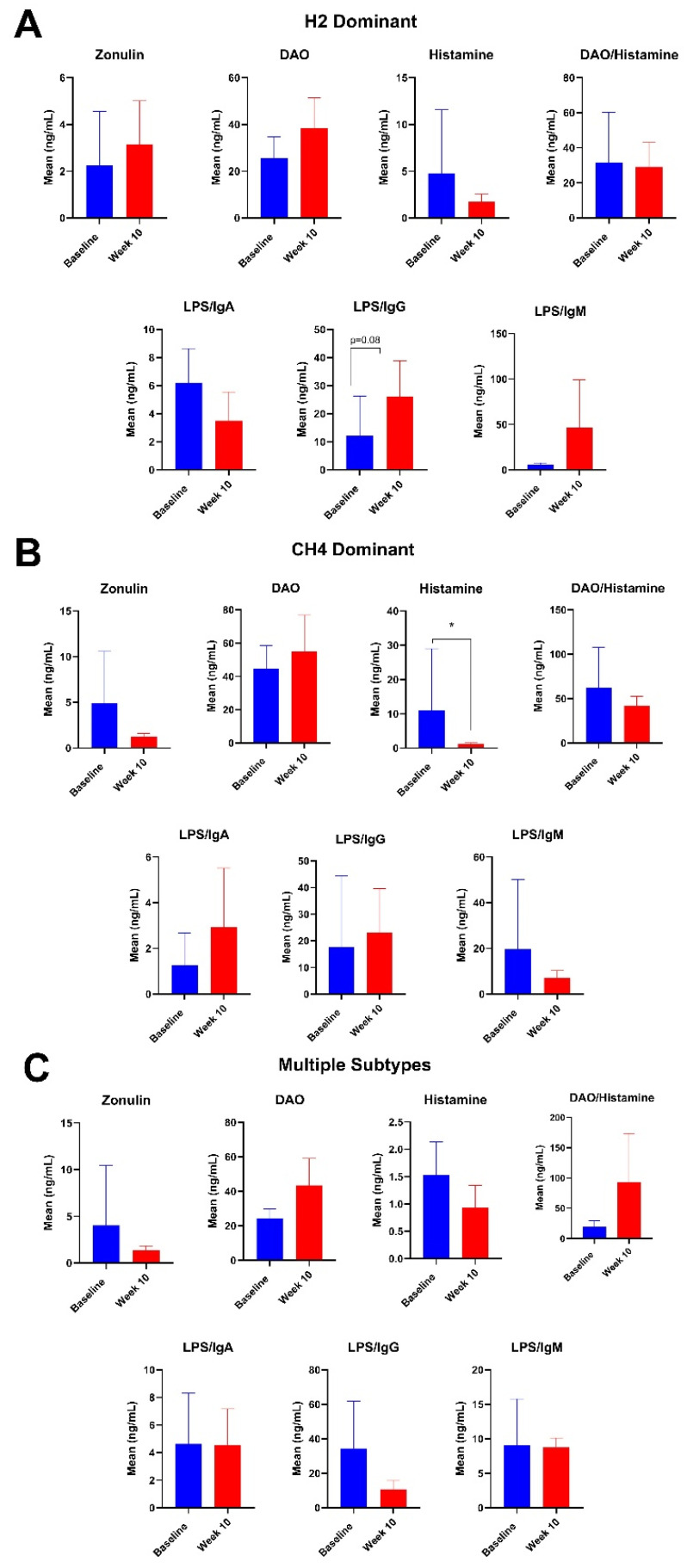
Mean plasma intestinal barrier function markers zonulin, DAO, histamine, DAO/histamine, LPS/IgA, LPS/IgG, and LPS/IgM for participants with (**A**) hydrogen-dominant SIBO, (**B**) methane-dominant SIBO, and (**C**) multiple subtypes of SIBO at baseline and week 10. * = *p* ≤ 0.05.

**Table 1 nutrients-16-03149-t001:** Supplement regimen during the study.

	Baseline to Week 10	Until Week 14
Biocidin liquid tincture	15 drops orally, morning and evening	15 drops orally, morning and evening
GI Detox+	2 capsules every evening, 1 h away from food	2 capsules every evening, 1 h away from food
Olivirex	-	2 capsules orally, morning and evening, with liquid tincture

**Table 2 nutrients-16-03149-t002:** SIBO breath test results for all participants at baseline.

Subject	Subtype
S01	H2
S02	H2
S03	H2
S04	H2
S05	CH4
S06	CH4
S07	H2, H2S
S10	CH4, H2S
S11	H2, CH4
S13	CH4
S14	H2, H2S
S15	H2
S16	H2
S17	CH4
S18	H2, H2S
S19	CH4

H2 = Hydrogen; CH4 = Methane, H2S = Hydrogen Sulfide.

**Table 3 nutrients-16-03149-t003:** Hydrogen, methane, and hydrogen sulfide breath test results.

Hydrogen SIBO				
Subject	Baseline	Week 6	Week 10	Week 14
S01	+	+	-	N/A
S02	+	-	+	LTFU
S03	+	+	LTFU	
S04	+	-	-	LTFU
S07 *	+	+	-	+
S08	+	Dropped after baseline		
S09	+	Dropped after baseline		
S11 *	+	-	LTFU	
S12	+	Dropped after baseline		
S14 *	+	Dropped after baseline		
S15	+	+	-	N/A
S16	+	pd	LTFU	
S18 *	+	+	+	+
**Methane SIBO**				
Subject	Baseline	Week 6	Week 10	Week 14
S05	+	+	+	-
S06	+	+	+	LTFU
S10 *	+	+	+	+
S11 *	+	+	LTFU	
S13	+	+	+	+
S17	+	pd	+	LTFU
S19	+	+	+	+
**Hydrogen Sulfide SIBO**				
Subject	Baseline	Week 6	Week 10	Week 14
S03	+	-	LTFU	LTFU
S07 *	+	-	+	-
S10 *	+	-	-	-
S14 *	+	Dropped after baseline		
S18 *	+	-	-	-

* Subject was positive for multiple subtypes of SIBO at baseline. LTFU = Lost to follow up; pd = Protocol deviation, subject did not complete breath test at visit; N/A = Not applicable, subject was a responder.

## Data Availability

The data presented in this study are available on request from the corresponding author.

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
