# Peer review of "An Oral Botanical Supplement Improves Small Intestinal Bacterial Overgrowth (SIBO) and Facial Redness: Results of an Open-Label Clinical Study"

_nutrients, 2024, doi:10.3390/nu16183149_

Round 1
Reviewer 1 Report
Comments and Suggestions for Authors
The authors address two very interesting topics, a GI issue (SIBO) in relation to a skin issue (erythrema) in a human study with an intervention which also seems to show an effect. The study design and execution looks good, the number of participants unfortunately is on the low side, but because of the novelty of this approach is still of interest and worthwhile publishing. There are however a few additions which would strengthen the manuscript. in the current version biomarkers (with respect to intestinal barrier and microbiome related) are only mentioned at a group. It would be interesting to analyze these in more detail by either are personalized approach or a subgroup based approach, such as comparing responder with non -responder. I realize the number of samples is too low to expect statistical significance, but also trends can be informative. Furthermore I am not able to open the supplementary Figures, so please try to provide these in a different format
Reviewer 2 Report
Comments and Suggestions for Authors
In abstract, clarify the number of participants and dropouts.
The structure of the abstract could be simplified to improve its flow. Consider breaking up longer sentences to make it easier to read.
In the introduction, discuss further what pathogenic bacteria are and why their increase is problematic, delving deeper into these points.
Regarding herbal alternatives, it would be useful to cite more studies demonstrating the efficacy of these therapies compared to conventional antibiotics. This would help strengthen the rationale for the study.
The possible limitations or challenges of the herbal approach should be important.
Please claoiry in methods, the "Products Used," "Composition of Formulas," "Dosage and Administration."
Highlight the route of administration at the beginning of the methodology.
It would be beneficial to include a brief justification for the selected ingredients, supported by studies that reinforce the choice of composition.
If available, add information about the credibility or certification of the commercial brand used in the study.
Include a section detailing the preparation and handling of the supplements to ensure the study's reproducibility.
Discuss the product's quality control, toxicity, ingredient purity, and possible contraindications.
Review botanical nomenclature, adding genus and species in italics for each plant, as some do not currently follow this format. Ensure the ingredients are described as clearly as possible.
Add information on how patient intake was monitored.
Round 2
Reviewer 1 Report
Comments and Suggestions for Authors
The authors did not do anything with my earlier remarks on the analysis of microbiota data and what is presented now on microbiota is very limited not to say too limited. Twenty to thirty percent increase in reads of selected species based on sequencing data which by nature are based on relative abundance does in fact not mean anything as goes for Firmicutes/Bacteroidetes ratios. Therefore I asked for working on relationship between microbiota and other biomarkers which was ignored. Another possibility would be to quantify specific species by quantitative PCR to show that quantitative changes are achieved upon intervention, but this now also lacking. In its current state the microbiome part of the study does not add anything and the manuscript would need revision.
Reviewer 2 Report
Comments and Suggestions for Authors
The manuscript may be now accepted.
